# Human Herpes Virus Genotype and Immunological Gene Expression Profile in Prostate Cancer with Prominent Inflammation

**DOI:** 10.3390/ijms26104945

**Published:** 2025-05-21

**Authors:** Elena Todorova, Anita Kavrakova, Goran Derimachkovski, Bilyana Georgieva, Feodor Odzhakov, Svitlana Bachurska, Ivan Terziev, Maria-Elena Boyadzhieva, Trifon Valkov, Elenko Popov, Chavdar Slavov, Ivan Tourtourikov, Vanyo Mitev, Albena Todorova

**Affiliations:** 1Department of Medical Chemistry and Biochemistry, MU-Sofia, 1431 Sofia, Bulgaria; e.todorova@genica.bg (E.T.); gueorguievab@yahoo.com (B.G.); itourtourikov@gmail.com (I.T.); mitev@medfac.acad.bg (V.M.); todorova_albena@abv.bg (A.T.); 2Genetic and Medico-Diagnostic Laboratory “Genica”, 1463 Sofia, Bulgaria; 3Department of Urology, St. Sofia Hospital, 1404 Sofia, Bulgaria; gderimachkovski@icloud.com; 4Department of General and Clinical Pathology, University Specialized Hospital for Oncology, 1756 Sofia, Bulgaria; feodor.odzhakov@gmail.com (F.O.); svitba@gmail.com (S.B.); 5Department of General and Clinical Pathology, UMHAT “Tsaritsa Ioanna—ISUL”, 1527 Sofia, Bulgaria; titia@abv.bg (I.T.); malena_96@abv.bg (M.-E.B.); 6Department of Infectious Diseases, MU-Sofia, Prof. Ivan Kirov Hospital, 1606 Sofia, Bulgaria; trifon_heim@abv.bg; 7Department of Urology, UMHAT “Tsaritsa Ioanna—ISUL”, MU-Sofia, 1527 Sofia, Bulgaria; ch.k.slavov@gmail.com

**Keywords:** prostate cancer, formalin-fixed and paraffin-embedded tissue, *Herpesviridae* family, intraprostatic inflammation, chronic urinary tract infection, risk stratification

## Abstract

We aim to investigate the role of the *Herpesviridae* family (*HHV*) in the onset and progression of prostate cancer (PCa) and to profile the local PCa immunological status. A total of 116 “tru-cut” biopsies (58 PCa and 58 benign prostatic hyperplasia [BPH]) and 49 formalin-fixed paraffin-embedded (FFPE) instances of PCa were analysed using real-time qPCR and histological examination. Infection with CMV, EBV, HHV6, and HHV7 was detected in 11.5% of the “tru-cut” biopsies (25.9% in BPH and 6.9% in the PCa group). In the formalin-fixed paraffin-embedded (FFPE) samples, infection was detected in 69.4% of the patients, with individual rates of EBV (47%), HHV6 (38%), HHV7 (41%), CMV (2.9%), HSV2 (2.9%), and VZV (5.8%). In the HHV-infected PCa cases, the histopathological landscape included intratumor lymphocyte infiltration with fibrosis and necrosis, periductal chronic inflammatory reaction and granulomatous lesions with foci of abscesses and necrosis, as well as inflammatory infiltration, chronic lymphadenitis, prostatic intraepithelial atrophy (PIA), and high-grade prostatic intraepithelial neoplasia (HGPIN). The majority of HHV-infected PCa patients were predominantly classified as grade G3/G4/G5 tumours, exhibiting perineural, perivascular, and lymphovascular invasion, seminal vesicle invasion, senile vesicle amyloidosis, and lymph node metastasis. Statistical analysis demonstrated a significant association between HHV infection and PCa (χ^2^ ≈ 20.3, df = 1, *p* < 0.0001; Fisher’s exact test, *p* < 0.0001) with an odds ratio of 6.50 (95% CI: 2.80–15.12). These findings suggest that long-term HHV infection could contribute to a complicated and potentially altered immune PCa tumour environment due to inflammation. This may serve as a predictor of aggressive disease progression.

## 1. Introduction

Prostate cancer (PCa) is the most frequently occurring solid organ neoplasm, the second most common type of cancer, and the fifth leading cause of death among men worldwide (GLOBOCAN data 2021) [1]. According to the latest data concerning PCa incidence and mortality [2], Bulgaria ranks among the most dramatically affected countries in Northern and Western Europe, North America, the Caribbean, Australia, New Zealand, and South Africa worldwide [2,3].

The lack of adequate national cancer screening programmes, as well as the complicated coronavirus disease (COVID-19) pandemic situation in recent years, leaves Bulgaria out of the cited favourable tendency to reduce the incidence of primary PCa. This can most likely be explained by the delayed diagnosis of clinically proven PCa cases, predominantly occurring at advanced stages. The optimal distinction between life-threatening and indolent PCa using molecular profiling and follow-up of silent intraprostatic infectious inflammation is currently defined as a serious challenge in clinical practice. Approximately 15% of human tumours in adults result from chronic untreated inflammatory conditions with an infectious aetiology, or otherwise from permanent exposure to harmful toxic factors [4,5].

Extensive molecular, histopathological, and epidemiological evidence supports the theory that persistent intraprostatic inflammation, caused by chronic urinary tract infection (UTI), is a crucial factor for viral-mediated genetic fluctuations, immune imbalance, and impaired apoptosis in infected cells, resulting in a patho-oncogenic cascade in PCa [6,7,8]. Furthermore, herpes viruses have sophisticated mechanisms for immune evasion, long latent persistence, and subsequent reactivation, some of which are officially recognised as carcinogenic factors [9].

Recent data indicate an association between viral infections with cytomegalovirus (CMV), Epstein–Barr virus (EBV), herpes simplex virus (HSV2), human herpesviruses (HHV6 and HHV7), and the malignant transformation of the prostatic epithelium [10,11]. To determine the prevalence and role of HHV in the pathophysiology of PCa onset and progression, we analysed a healthy control group (clinically free of PCa), as determined by seminal fluid status, a clinical control group (BPH), and a clinical group with PCa (solid tumours) of Bulgarian origin. We further explored the presence of co-infections, specifically EBV combined with HHV6/7 in prostatic tumorigenesis and progression.

The current study contributes additional evidence that supports the modern hypothesis regarding the role of persistent HHV infection as a potential etiological co-factor for intraprostatic inflammation, resulting in PCa tumorigenesis, triggering a complicated and potentially altered immune tumour profile. To assess the PCa tumour microenvironment for impaired immunity, we also performed a pilot RNA-expression analysis for PCa-associated immunological factors (IL1β, IL10, IL18, TNF-α, TLR4, GATA3, CD68).

## 2. Results

### 2.1. Molecular Virological Results

Real-time PCR performed on extracted DNA detected infections with CMV, EBV, HHV6, and HHV7 in 11.5% of the patients with “tru-cut” biopsies (25.9% in Benign prostatic hyperplasia, BPH, and 6.9% in PCa group). Infections with EBV, HHV6, HHV7, CMV, HSV2, and VZV were detected in 69.4% of the FFPE-tested patients (Table 1). The prevalence in the HHV-positive PCa cases was 47% (EBV), 38% (HHV6), 41% (HHV7), 2.9% (CMV), 2.9% (HSV2), and 5.8% (VZV), with the most prevalent co-infections being EBV/HHV6/HHV7. All positive samples amplified ≥5 cycles earlier than the kit’s diagnostic cut-off (mean Ct 29 ± 1.2). No differentiation could be made for the virus residing in tumour epithelial cells, infiltrating leucocytes, or stromal elements based on the currently used methods.

Moreover, we found a significant proportion of viral co-infections (34/49 HHV-positive PCa formalin-fixed paraffin-embedded [FFPE] samples). EBV was the predominant factor in the most-frequently detected co-infections: EBV/HHV7 (31%), EBV/HHV6 (23%), EBV/CMV (8%), EBV/VZV/HHV7 (8%), and EBV/HSV2 (7%). In the current study, a combined co-infection variant of HHV6/HHV7 was also reported in 23% of the cohort (Figure 1).

The estimated frequency of HHV infection in the healthy control group was 6%.

### 2.2. Histopathological Results

In all 15 BPH biopsies positive for HHV viral infection, the predominant morpho-histological findings indicative of an inflammatory process included atypical, atrophic, and precancerous changes, as well as inflammatory infiltration, amyloid body formations, and a cytopathic effect.

In 34 FFPE HHV-infected tumour tissues, histopathological examination revealed intratumoural lymphocyte infiltration with fibrosis and necrosis, periductal chronic inflammatory reaction, granulomatous lesions with foci of abscesses and necrosis, inflammatory infiltration, chronic lymphadenitis, PIA, and high-grade prostatic intraepithelial neoplasia (HGPIN).

Most virus-infected (32/34) FFPE samples were classified as poorly or moderately differentiated prostatic tumours corresponding to G3, G4, and G5 by Gleason. Histological features of aggressiveness, poor prognosis, and disease progression, such as perineural and/or perivascular invasion, lymphovascular invasion, seminal vesicle invasion, senile vesicle amyloidosis, and lymph node metastasis, were found across all samples (Figure 2). The HHV may be present within the prostate tissue or may have infiltrated mononuclear cells.

Massive inflammatory processes with foci of high-grade prostatic intraepithelial neoplasia (HGPIN) lesions were observed in a negligible proportion (2/34) of well-differentiated virus-infected FFPE tumour samples corresponding to the G1 prognostic group, and findings of a massive inflammatory process with the foci of HGPIN lesions were observed. All virus-negative FFPE samples were identified as well-differentiated G1 prostatic tumours without chronic inflammation or tumour invasion.

### 2.3. Pilot Immunological Results

To investigate the immunological profile in the PCa cohort, the ImmunoQuantex molecular assay was used to determine IL-1b, IL-10, IL-18, TNFα, TLR4, GATA3, CD68, and B2M (as a housekeeping gene) levels. Immunological factor ratios (TLR/GATA3; TNF-α/IL18; IL10/IL18; 1L1B/CD68) were used to determine an inflammation index based on the manufacturer’s instructions, with a highly elevated inflammatory profile determined as an inflammation index > 50 with a cut-off of 50%. In total, 61.8% (21/34) of FFPE PCa samples positive for HHVs were above the cut-off, with 85.7% (18) of patients presenting with an extremely high inflammatory index (>80%), indicating massive intraprostatic inflammation (Table 2).

Fisher’s exact test was used to investigate the association between HHV positivity and the inflammation profile for each inflammation index threshold. When comparing HHV-positive vs. -negative samples with the 50% cut-off, HHV-positive samples showed a statistically significant association in terms of inflammation levels (*p* = 2.59 × 10^−5^). With a cut-off of 80%, the association was again statistically significant (*p* = 2.46 × 10^−4^). All of the examined FFPE PCa samples with a highly elevated inflammatory index corresponded to a predominant profile of a more aggressive and progressive histopathological picture (G3/G4/G5), including massive inflammation. This observation was further supported by the strongly aberrant and increased expression levels of IL1β, IL18, TNF-α, and CD68. In contrast, BPH samples did not have an elevated inflammatory index above the cut-off value. The expression profiles of IL1β, IL10, IL18, TNF-α, TLR4, GATA3, and CD68 in HHV-infected PCa FFPE biopsies indicated an inflammatory environment conducive to immune alteration and potential tumour progression.

## 3. Discussion

Human herpesviruses can influence tumour biology indirectly by sustaining chronic inflammation [12] and by modulating local immune responses [13]. Established examples include EBV, which is causally linked to Burkitt lymphoma and nasopharyngeal carcinoma [14] and HHV-8 in Kaposi sarcoma [15]. Observational studies have suggested associations between EBV or HHV-8 infection and prostate cancer risk, with some cohorts reporting higher odds in seropositive men or an elevated risk with increased EBV antibody titres [16,17,18].

Accumulating evidence indicates that persistent HHV infection can remodel the tumour microenvironment via cytokine-driven inflammation and immune cell recruitment. Specifically, EBV latent genes and the HHV-8-encoded viral IL6 potentiate NFkB/STAT3 signalling, boosting IL6, CCL2, and allied chemokines that attract monocytes and myeloid-derived suppressor cells [19,20]. Moreover, the herpesviruses’ latency is often associated with immunosenescence [21,22], which could also be crucial for tumorigenesis. We report indirect pilot data on herpesvirus-mediated altered immune function and reactive response, in terms of immune cell infiltration and elevated inflammation in the tumour microenvironment (Figure 2).

We estimated the frequency of EBV, HHV6, HHV7, CMV, HSV2, and VZV infections to be 69.4% of all PCa tested (FFPE), distributed as follows in the positive samples: 47%, 38%, 41%, 2.9%, 2.9%, and 5.8%, respectively. In contrast, we found CMV, EBV, HHV6, and HHV7 infection in only 11.5% of the patients with “tru-cut” biopsies (25,9% in BPH and 6.9% in the PCa group). We attributed the significantly higher frequency of HHVs to the optimisation of FFPE sample selection by pre-labelling the tumour content. The estimated higher number of HHVs in BPH was anticipated because of the observed signs of early precancerous changes accompanied by prominent inflammation. We expect that future follow-up will reveal that a high proportion of these patients could develop both BPH and PCa, a combination frequently reported [23].

Based on the literature, EBV occurs in ~8–9% (BPH) and ~5–37% (BPH/PCa) of HGPIN and PCa cases [24,25]. Furthermore, there is a positive association between EBV and PCa across all histological grades, from well-differentiated to hypernephroid or solid tumours, with an unfavourable prognosis according to the Gleason classification [26]. The results of our study align with previous reports, with 32% of all tested FFPE samples and 47% of all HHV-infected probands presenting with an EBV infection.

Moreover, we found a significant viral co-infection proportion (38%) in FFPE samples with EBV as the predominant participant in the most frequent combinations: EBV/HHV7 (31%), EBV/HHV6 (23%), EBV/CMV (8%), EBV, VZV, HHV7 (8%), and EBV/HSV2 (7%). A variant of co-infection with HHV6/HHV7 (23%) was also detected. We hypothesise a mechanism of cooperative virus-mediated tumorigenic activity, where, in the presented cases of prostate neoplasia, EBV works in tandem with other *Herpesviridae* members, mainly HHV-6/7, to promote tumorigenesis and stimulate malignant cell proliferation (Figure 1). Our data are in concordance with the literature describing cooperative contributions to prostatic oncogenesis by EBV, HPV, HHV6, and other viruses [27]. Our results also suggest that EBV represents a potential etiological factor not only limited to lymphoepithelioma-like carcinomas, but that it also plays a possible role in provoking PCa tumorogenesis and progression, PCa/BPH, and high-grade dysplasia preceding PCa [24,28,29]. Localization of EBV within the tumour, either within prostate tumour cells or the lymphocytic infiltration, suggests a role for the virus in tumour promotion. In the current study, we assume the possibility of independent or combined HHV6 (38%) activity in the presented variants of co-infection as a virus with a role in prostate tumourigenesis. HHV6-induced inhibition of the key nuclear localisation and functions of p53 has been experimentally proven [30]. The described events lead to uncontrolled growth and apoptosis evasion by infected cells and provide us a reason for future investigations of the impact of HHV6 on clinical PCa behaviour. The literature clarifies and describes the potential for HHV6 action to enhance tumorigenicity in detail; however, data regarding its active prevalence and functional role in PCa pathogenesis are scarce. Unfortunately, data regarding the potential oncogenic role of HHV7 in PCa are insufficient, despite the significant proportion detected in the current study (41%) showing at least the presumptive impact of HHV7.

Infection with CMV in prostatic epithelial cells has been reported to trigger mutagenesis and angiogenesis by inducing cell cycle progression, the activation of cell motility and migration, *VEGF* expression induction, DNA damage repair inhibition, and chromosomal aberrations, leading to the inhibition of apoptotic pathways [31,32,33]. The role of CMV has been reported mostly in the precancerous and/or initial PCa stages, compared to a markedly lower activity in prostate tumour cells, similar to HPVs [34]. In the current study, CMV was detected in only one case with moderately differentiated PCa and an aggressive profile, which is in contrast to the significantly higher frequencies reported in the literature.

There are data regarding the increased risk of PCa development and positive association of HSV1/2 with prostatic oncogenesis based on elevated HSV1/HSV2 miRNAs in the PCa profile compared to BPH [35]. Meta-analyses of tumour tissues have revealed a positive association between HSV2 and PCa in terms of serological prevalence [36,37]. Our data from FFPE PCa cases revealed only sporadic HSV2-infected cases and no HSV1-infected cases. According to the literature, there is a positive association between VZV and an increased risk of PCa onset; however, further investigation is needed [38].

Regarding HHV8, all tested FFPE PCa samples were negative. This finding is not surprising, because our patients were of Bulgarian origin. Our country is not an endemic or subendemic region for Kaposi sarcoma, reported often to have a positive association with an increased risk of PCa development or even an aggressive PCa subtype.

We describe not only an intensive inflammatory process in the studied PCa FFPE samples, but also an indisputable correlation between HHV infection and the presence of poorly and moderately differentiated prostate tumours (G3,4,5) with prominent histological features of aggressive clinical behaviour and metastatic potential (see the histological results) (Figure 2) Moreover, we categorised persistent HHV infection (69.4%) as an unfavourable prognostic bioindicator not only for PCa onset, but also for PCa progression, as has already been mentioned [39,40]. Our results support the literature, where a positive HHV infection was considered to be an important criterion for proper risk stratification, relapse of PCa, and even a medium–short life expectancy of patients/a 5-year survival without relapse after radical prostatectomy (RP) in infected/uninfected men (38.1/96.3%) (*p* < 0.05), or medium–short life expectancy in HHV-infected/uninfected patients to combination treatment 27.71/76.6 months (*p* < 0.05) [41]. The evidence for HHV-mediated PCa progression was confirmed by abundant EBV-infected cells registered in cases of histo-architecturally distorted prostate carcinoma removed after androgen deprivation therapy [24]. The cited data on the linkage between EBV and more aggressive forms of PCa coincide with the data reported in the current study [42].

HHV infection could be stimulated by the hormonal/ligand effects provoked by unbalanced androgen homeostasis, such as those that were not an exception in the presented PCa cases with a CRPC (Castration-Resistant Prostate Cancer) profile. This thesis is theoretically supported by the cited evidence about the presence of receptors on prostate cells for HHV entry [43]. Reporting a massive inflammatory process and HGPIN in a negligible series well-differentiated infected FFPE, and totally absent inflammation and tumour invasion in uninfected ones, we hypothesize the participation of HHVs in precancerous changes and initial PCa onset.

By analysing IL1β, IL10, IL18, TNF-α, TLR4, GATA3, and CD68 expression in 50% of PCa HHV-infected FFPE with an elevated inflammation index (61.8%/21), we detected hyper-expressed levels of IL1β, IL18, and TNF-α in the tumour microenvironment. Similar fluctuating levels are cited in association with the elevated intensity of neoplastic progression, angiogenesis, metastases, and lymph node metastases, mainly to avoid antitumour responses [44,45,46]. IL-1β [44,47], IL-18 [48,49,50,51], and TNF-α [52] are known to be associated with tumour invasion, progression, and metastatic potential. The presence of human herpesviruses (HHVs) within the PCa tumour microenvironment suggests an altered and complicated local immune status, where, likely due to viral infection, the overall local immune control appears to be impaired. The predominance of ab an elevated inflammation index in the studied HHV-infected PCa group correlates with histological findings on intraprostatic inflammation. The reported data suggest that the proven inflammatory process can contribute toward tumour onset and also possibly progression in prostate cancer. Further research is required to elucidate these effects comprehensively.

Significantly lower frequencies of HHVs were detected in the healthy controls, which was not surprising because our study was focused mainly on the impact of viral infections on the precancerous prostate and/or PCa. There is no doubt that many latent viral factors can be reactivated in the tumour-affected prostate gland. However, our promising results, in combination with the cited data concerning the total absence of HHVs in other tumour types (even epithelial), once again confirm the hypothesis regarding the contribution of HHVs in tumorigenicity in the prostate, probably triggering PCa onset and future progression. Given this evidence, future research should focus on integrating HHV screening into prostate cancer risk assessments, particularly in patients presenting with high-grade disease or idiopathic intraprostatic inflammation. Longitudinal studies assessing the impact of HHV status on disease progression, treatment response, and patient outcomes will be critical to determining whether viral profiling should be incorporated into routine clinical practice. Ultimately, understanding the role of HHVs in prostate cancer pathogenesis could lead to novel biomarker-driven approaches for personalised treatment strategies.

This study has several important limitations. First, the cohort size is constrained by the small Bulgarian population and by the fact that total prostatectomy is rarely performed in older, comorbid men, who most often present with prostate cancer. Second, we were unable to pinpoint whether EBV DNA was located in the tumour epithelial cells or in the infiltrating lymphocytes because immunohistochemical localisation was not feasible for the current series of samples. Finally, the immunological profile provides only an overview of the PCa microenvironment. Future work should quantify individual immune-factor expression in HHV-positive tumours to clarify the mechanistic risks. Despite these constraints, our study supports the role of human herpes virus infection as a co-factor in the initiation and progression of prostate cancer and should prompt further, larger-scale studies.

## 4. Materials and Methods

### 4.1. Patients and Samples

Two groups of Bulgarian PCa patients’ samples were selected for the current study: 116 “tru-cut” biopsies (PCa 58/BPH 58) and 49 FFPE tissues preliminarily marked for tumour target zones (Table 3). The tru-cut biopsy technique was used to obtain 12–14 core tissue samples from each patient per diagnostic standard, with an extra sample for the purposes of this study. BPH patients were selected for the presence of BPH and absence of any chronic, metabolic, or other comorbidities. In addition, a control group of 100 healthy controls with non-invasive samples (semen liquid) representative for prostatic exprimate content were included in the study. The healthy status was confirmed by standard urological examination and prostate-specific antigen (PSA) investigation. This control group was used as a reference for HHV infection status in clinically healthy patients free of PCa. Written informed consent was obtained from all Bulgarian probands (average age 67.2 years) before molecular virological testing.

### 4.2. Nucleic Acid Extraction

Total DNA from tru-cut biopsy samples and semen liquid was extracted using DNA isolation kit AmpliSens, Ecoli s.r.o, Slovak Republic. Total DNA/RNA from FFPE samples was extracted using the QIAamp DNA FFPE Tissue Kit and RNeasy DSP FFPE Kit (QIAGEN, Hilden, Germany), following the manufacturer’s instructions. To ensure the quality of high-molecular-weight RNA extracted from these samples, we measured RNA concentration and purity using a Nanodrop spectrophotometer. Several steps for the pre-analytical preparation of the semen liquid were performed. To the ejaculate sample (300 µL), 60 µL 1 M dithiothreitol (DTT) and 20 µL proteinase K were added. The samples were centrifuged for 15 min at 2500 rpm at 4 °C. The supernatant was carefully discarded without disturbing the sediment. The pellet was resuspended in 200 µL ice-cold buffered NaCl. The resuspended sediment was transferred to a sterile 1.5 mL centrifuge tube ready for further use. The quality of RNA extraction and cDNA synthesis was verified via internal control metrics available within the reagent kits used in this study.

### 4.3. Real-Time PCR

Molecular analyses for HHV infection, detection, and expression profile of immunological factors (IL1β, IL10, IL18, TNF-α, TLR4, GATA3, CD68) were performed using commercial diagnostic kits: the SensiFAST cDNA Synthesis Kit RXN BIOLINE, ImmunoQuantex C/V REAL-TIME PCR Genotyping Kit(DNA Technology Moscow, Russia), and AmpliSens amplification kits (Ecoli s.r.o, Bratislava, Slovak Republic). All PCR amplifications for HHV genotyping and immunoprofiling were performed according to the manufacturer’s instructions.

The type-specific DNA genome detection of HHVs was performed under the following real-time PCR conditions: initial denaturation 95 °C/15 min., 95 °C/5 s; 60 °C/20 s; 72 °C/15 s × 5 cycles, 95 °C/5 s; 60 °C/30 s; 72 °C/15 s × 40 cycles 4 °C storage. Target regions for the amplification of HHV viruses were focused on the genes essential for the viral replication (kit provided: EBV—*LMP* gene, Ct limit 35; CMV—exon 4 of *MIE* gene, Ct limit 36; HHV6—DNA polymerase catalytic subunit, Ct limit 36.). All patient samples were run in triplicate.

The following real-time PCR conditions were applied for the target *IL1β*, *IL10*, *IL18, TNF-α*, *TLR4*, *GATA3,* and *CD68* RNA expression measurements: pre-initial denaturation 80 °C/30 s; initial denaturation 94 °C/1.30 min.; 94 °C/30 s; 64 °C/15 s × 5 cycles; 94 °C/10 s; 64 °C/15 s × 45 cycles, 94 °C/5 s and 10 °C storage. This stage was preceded by a reverse-transcription step. The applied house-keeping gene in the commercial kit for RNA expression immunological profiling was *B2M*. A secondary housekeeping gene PSA was used with the purpose of proving, with exceptional fidelity, the presence of target prostatic cells in the analysed clinical samples and the quality of cDNA synthesis. Different PSA-specific primers were used for PCR: forward, 5′-AGCATTGAACCAGAGGAGTTCT-3′ (nucleotides 4024–4042 of exon 3 and 4186–4188 of exon 4 of the PSA gene, GenBank #M27274) and reverse, 5′-CCCGAGCAG GTGCTTTTG-3′ (nucleotides 4307–4322 of exon 4 and 5699–5700 of exon 5 of the PSA gene), as described by Hessels et al. [53]. The following PCR conditions were applied: master mix with final volume 10 μL: 2 μL cDNA input; 0.4 μL of specific PSA primers with a concentration 20 pmol/μL; 1 μL ddNTP (5 mM) and 1 μL 10X Buffer; 0.05 μL Taq DNA Polymerase 250 U 5 U/μL (GeNet Bio, Yuseong-gu, Daejeon, Republic of Korea) and 5.15 μL PCR water. The PCR cycling conditions were as follows: initial denaturation 95 °C/10 min, followed by 95 °C/30 s; 60 °C/75 s; 72 °C/60 s × 42 cycles.

## 5. Conclusions

Our experimental results support an association between local herpesvirus infection, inflammatory signalling, and aggressive pathological features. Whether antiviral treatment or the suppression of herpesvirus activity can alter prostate cancer risk and progression remains an open question; further prospective studies that integrate virological stratification with clinical outcomes and therapeutic interventions are required to investigate the clinical value of added targeted antiviral strategies to standard prostate cancer care.

## Figures and Tables

**Figure 1 ijms-26-04945-f001:**
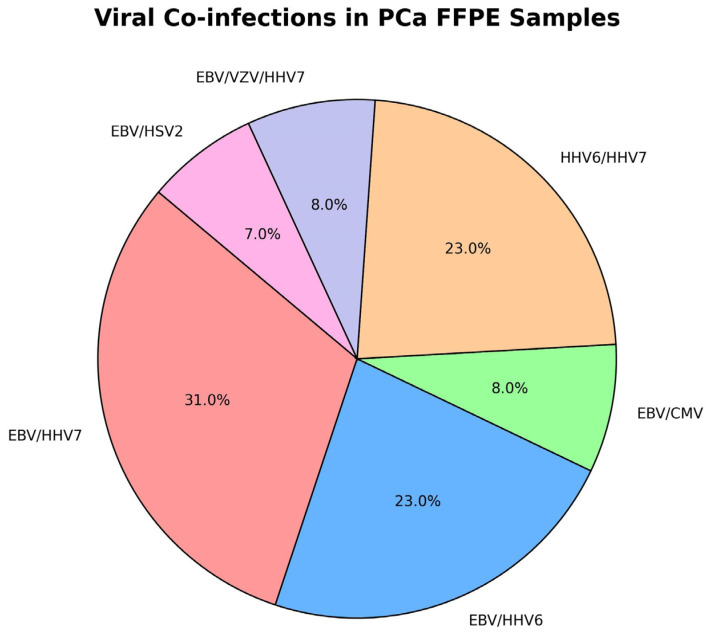
Viral combinations in the co-infected FFPE group. To further investigate HHV infection in PCa, we compared infection rates for 49 patients with PCa FFPE biopsies and 58 tru-cut BPH biopsies (Table 1). Chi-square test revealed a significant association with HHV infection, with 69.4% prevalence in the PCa group (χ^2^ ≈ 20.3, df = 1, *p* < 0.0001) compared to 25.9% in the BPH group. Fisher’s exact test confirmed a highly significant association between HHV infection and prostate cancer (*p* < 0.0001). The odds of HHV positivity are significantly higher in patients with PCa compared to those with BPH, with an odds ratio of 6.50 (95% CI: 2.80–15.12).

**Figure 2 ijms-26-04945-f002:**
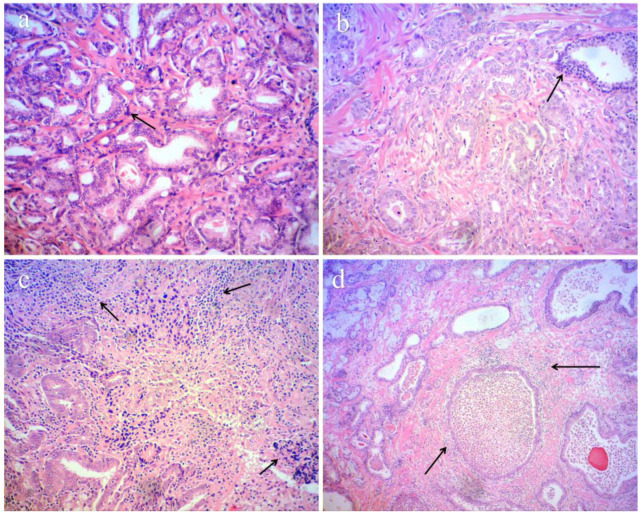
Histo-pathological findings of massive intraprostatic inflammation and aggressiveness in infected prostate acinar adenocarcinoma cases. (**a**) GS 3 + 4 = 7, WHO grade group 2; intratumor lymphocyte infiltration, (**b**) GS 3 + 4 = 7, WHO grade group 2; periductal chronic inflammatory reaction; senile vesicular amyloidosis, (**c**) GS 5 + 4 = 9, WHO grade group 5; intratumor lymphocyte infiltration with fibrosis and necrosis; perineural, perivascular, lymphovascular invasion, and lymph node metastasis. (**d**) GS 3 + 4 = 7, WHO grade group 2; granulomatous inflammation with foci of abscesses and necrosis; seminal vesicle invasion.

**Table 1 ijms-26-04945-t001:** Prevalence of HHV infections in patients with BPH and PCa.

Group	Number	Sample Type	HHV +	HHV −	Group Information
Prostate Cancer	58	Tru-cut Biopsies	4/6.9%	54/93.1%	Randomly selected biopsies (from core samples)
Benign Prostatic Hyperplasia	58	Tru-cut Biopsies	15/25.9%	43/74.1	Randomly selected biopsies (from core samples)
Prostate Cancer	49	FFPE Tissue (Tumour-Targeted)	34/69.4%	15/30.6%	Areas pre-identified as tumour target zones
Healthy Controls (Clinically PCa-Free)	100	Semen fluid	6/6%	94/94%	Clinically healthy subjects were tested for HHV status

**Table 2 ijms-26-04945-t002:** Inflammation index status in HHV-positive/negative FFPE PCa samples.

PCa + ve FFPE Samples	Inflammation Index <50%	Inflammation Index >50% and <80%	Inflammation Index >80%
*HHVs* positive	13/34	3/34	18/34
*HHVs* negative	15/15	0/15	0/15

**Table 3 ijms-26-04945-t003:** Patient male cohort: demographics and anthropometrics.

Group	Group Information	Number	Age (Years)	PSA (ng/mL)	Origin/Ethnic Race
Prostate Cancer	Randomly selected biopsies (from core samples)	58	68.7 ± 5.09	17.06 ± 15.84	Bulgarian/Caucasian
Benign Prostatic Hyperplasia	Randomly selected biopsies (from core samples)	58	64.6 ± 6.9	9.3 ± 1.9	Bulgarian/Caucasian
Prostate Cancer (FFPE)	Areas pre-identified as tumour target zones	49	66.6 ± 9.6	63.3 ± 35.6	Bulgarian/Caucasian
Healthy Controls (Clinically PCa-Free)	Clinically healthy subjects were tested for HHV status	100	60.1 ± 7.7	All below <3 ng/mL	Bulgarian/Caucasian

## Data Availability

Data are contained within the article. Further inquiries can be directed to the corresponding author.

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
