# Peer review of "Human Herpes Virus Genotype and Immunological Gene Expression Profile in Prostate Cancer with Prominent Inflammation"

_ijms, 2025, doi:10.3390/ijms26104945_

Round 1
Reviewer 1 Report (Previous Reviewer 3)
Comments and Suggestions for Authors
The authors have addressed my concerns in a satisfactory manner. I now agree with publication of this manuscript.
Author Response
Thank You so much for the positive assessment and the significant help to improve our manuscript through the insights in your comment
Reviewer 2 Report (Previous Reviewer 2)
Comments and Suggestions for Authors
The authors investigated the association between Herpesviridae family (HHV) infections and prostate cancer (PCa) by analyzing 116 “tru-cut” biopsies and 49 FFPE samples. They identified active HHV infections in both BPH and PCa cases, with a higher prevalence in FFPE samples. Histopathological analysis of HHV-infected PCa cases revealed significant inflammatory and malignant features, including high-grade tumors and metastatic progression. Statistical analysis confirmed a strong association between HHV infection and PCa, suggesting that chronic HHV infection may contribute to an immunocompromised tumor environment and predict aggressive disease progression. I have few questions
- Could authors clarify why seminal fluid was used as a control group instead of other sample types like blood or urine?
- What criteria were used to select the healthy controls, and how was their prostate cancer-free status confirmed?
- The authors used FFPE tissue for viral detection. Can you elaborate on how the preservation method might impact viral DNA/RNA detection sensitivity?
- The prevalence of viral co-infections was significantly higher in PCa patients. How did the study ensure accurate differentiation between latent and active infections?
- The histopathological findings showed signs of inflammation and tumor aggressiveness. Could these inflammatory markers be a result of other underlying conditions unrelated to HHV infection?
- Can the authors provide a detailed breakdown of the statistical methods used for significance testing and justify why these were appropriate?
- Elevated cytokines like IL-1β, IL-18, and TNF-α were detected. Could there be a feedback loop where viral infection induces cytokine expression, promoting tumorigenesis? Did the study explore correlations between cytokine levels and clinical outcomes?
- BPH samples showed higher HHV presence, attributed to precancerous inflammation. Were other confounding factors, like age or comorbidities, evaluated to rule out non-specific inflammation?
- Consider clarifying the use of "reactive response" and "immune compromise" within the tumor microenvironment. Are these based on immune cell infiltration, cytokine profiling, or other biomarkers? Consistent terminology will enhance clarity.
Author Response
Below is our point‐by‐point response to the reviewers’ comments. We thank the reviewers for their constructive suggestions, which have helped us improve the clarity, depth, and presentation of our manuscript. Our responses to each comment are provided below the original remark.
Answers Reviewer 2
The authors investigated the association between Herpesviridae family (HHV) infections and prostate cancer (PCa) by analyzing 116 “tru-cut” biopsies and 49 FFPE samples. They identified active HHV infections in both BPH and PCa cases, with a higher prevalence in FFPE samples. Histopathological analysis of HHV-infected PCa cases revealed significant inflammatory and malignant features, including high-grade tumors and metastatic progression. Statistical analysis confirmed a strong association between HHV infection and PCa, suggesting that chronic HHV infection may contribute to an immunocompromised tumor environment and predict aggressive disease progression. I have few questions
- Could authors clarify why seminal fluid was used as a control group instead of other sample types like blood or urine?
Thank you for this question. We chose seminal fluid as a healthy control group marker because it is the only non-invasive biological sample from a healthy male, but representative of the prostate. Our aim was to show results for clinically healthy subjects tested for HHV status. A blood or urine sample does not contain prostate exprimate that could be used for prostate investigation.
- What criteria were used to select the healthy controls, and how was their prostate cancer-free status confirmed?
Thank you for your observation. The healthy status free of PCa diagnosis was confirmed by standard urological examination and PSA investigation (normal values). We have added this to the materials and methods section
- The authors used FFPE tissue for viral detection. Can you elaborate on how the preservation method might impact viral DNA/RNA detection sensitivity?
Thank you for this question. As a powerful source of data, cancer FFPE samples are used for RNA extraction and consequent analyses worldwide. To guarantee the quality of extracted high-molecular weight RNA/DNA inputs, each sample underwent subsequent total RNA/DNA concentration and purity measurements by nanodrop machine. While formalin fixation causes extensive crosslinking between proteins and nucleic acids and leads to fragmentation and chemical modifications of viral DNA/RNA, which can lead to subsequent lower viral detection compared to fresh or frozen tissues, this is a limitation widely known in the field. The applied commercial kits for RNA FFPE extraction, reverse transcription and RNA expression measurements (see Chapter “Material and methods”) are widely used in the field as their quality guarantees a high-yield nucleic acid extraction and subsequent detection.
- The prevalence of viral co-infections was significantly higher in PCa patients. How did the study ensure accurate differentiation between latent and active infections?
Thank you for this observation. The kits used for HHV detection in the manuscript Chapter 4.2. detect only active, i.e. productive HHVs infection with no detection for borderline or latent stages of HHVs infection. The assays have a predefined detection threshold that excludes the low-level viral genome copies typically seen in latent infections, meaning that viral load measurements correspond exclusively to active infections. Furthermore, the tissue areas chosen for FFPE processing were pre-identified as tumor target zones, ensuring that the detected viral signal correlated with the pathologically relevant, active infection rather than incidental or latent viral presence.
- The histopathological findings showed signs of inflammation and tumor aggressiveness. Could these inflammatory markers be a result of other underlying conditions unrelated to HHV infection?
Thank you for this question. Although inflammation and aggressive tumor features can arise from various underlying conditions, we controlled for this possibility by comparing HHV-positive prostate cancer samples to both benign prostatic hyperplasia and clinically healthy control samples. Inflammatory response in all the cohort groups has been evaluated in terms of morphology and degree of presentation. Comparison within PCa group has shown significant increase in the extent of inflammation in those positive for HHV. Comparison between PCa group positive for HHV and healthy control group has shown significantly higher extent of inflammatory response. Similar findings have been observed in terms of PCa group positive for HHV compared to PCa group negative for HHV. These circumstances have allowed to draw a conclusion that there is a direct correlation between HHV infection and degree of inflammatory response. The HHV-positive FFPE samples consistently showed a markedly elevated inflammatory profile and specific histopathological features—such as intratumoral lymphocyte infiltration, fibrosis, and necrosis—that were not present in HHV-negative cases. Of course, comorbidity due to other underlying conditions not associated with viral inflammation cannot be excluded, but this information was not made specifically available for individual patients.
- Can the authors provide a detailed breakdown of the statistical methods used for significance testing and justify why these were appropriate?
Thank you for the question. We used a Chi-squared test to compare infection rates between the PCa FFPE group and the BPH group, followed by a Fisher’s exact test to confirm the association due to the sample size. Finally, the odds ratio was calculated to quantify the strength of the association between HHV carriers and PCa. These methods are appropriate for our study as the data is formed from categorical variables (presence/absence of HHV infection vs disease group classification), which fits the assumptions for both chi-square and Fisher’s exact tests. The odds ratio with confidence intervals adds clarity to the magnitude and reliability of the association.
- Elevated cytokines like IL-1β, IL-18, and TNF-α were detected. Could there be a feedback loop where viral infection induces cytokine expression, promoting tumorigenesis? Did the study explore correlations between cytokine levels and clinical outcomes?
Thank you for the great question. These cytokines are known to promote chronic inflammation and have been implicated in tumorigenesis, suggesting that the viral infection could indeed stimulate cytokine expression, which in turn may promote further tumor progression. While we observed that the elevated inflammatory index (based on the cytokine levels and additional immunological markers) was predominantly found in more aggressive tumors, we did not explore these correlations as we need further samples to accurately correlate these cytokine levels with clinical endpoints.
- BPH samples showed higher HHV presence, attributed to precancerous inflammation. Were other confounding factors, like age or comorbidities, evaluated to rule out non-specific inflammation?
Thank you for this question. BPH patients were selected to be healthy patients with BPH without accompanying chronic, metabolic and alternative diseases. We have included this in the materials and methods section for clarity.
- Consider clarifying the use of "reactive response" and "immune compromise" within the tumor microenvironment. Are these based on immune cell infiltration, cytokine profiling, or other biomarkers? Consistent terminology will enhance clarity.
Thank you for this observation. In our study, “reactive response” refers to the active inflammatory reaction observed in the tumor microenvironment—this includes both immune cell infiltration (for example, the presence of intratumoral lymphocytes) and elevated cytokine levels (notably IL-1β, IL-18, and TNF-α) detected via qPCR. In contrast, “immune compromise” describes a scenario where, despite the reactive inflammatory signals, the overall local immune control appears impaired—suggested by an imbalance in cytokine ratios and the histopathological evidence of aggressive tumor features (such as perineural and lymphovascular invasion). We have clarified this in the manuscript.
We hope that these revisions and clarifications adequately address the reviewers’ concerns and improve the manuscript. Thank you again for your valuable input.
Reviewer 3 Report (Previous Reviewer 1)
Comments and Suggestions for Authors
In the present work Todorova et al. reported on additional evidence supporting the modern hypothesis regarding the role of persistent HHV infection as a potential etiological co-factor for intraprostatic inflammation, resulting in Prostate Cancer (PCa) tumorigenesis, triggering an immunocompromised tumour profile. In order to evaluate their hypothesis, the authors assessed the PCa tumor microenvironment for impaired immunity, while performing a pilot RNA- expression analysis for PCa-associated immunological factors, some of them including IL1β, IL10, IL 18, TNF-α, TLR4, GATA3, CD68. Although the idea is interesting there are several important aspects that need to be addressed. First of all, the authors report on DNA and RNA extraction from FFPE samples, yet my first concern is that FFPE samples are very hard to obtain RNA from. How did the authors manage to overcome RNA cleavage, after preserning samples in paraffin? The authors mention the evaluation of several genes, but there are no primers mentioned for all these genes. In addition, there should be two different sections; one for DNA, RNA extraction and one for qRT-pCR. Further on, how was viral load estimated? In order to establish a correlation between viral infection and PCa, it is imperative to measure the levels of viral genome in a sample. Yet, there are no data presented for those questions. In particular, the authors did not present any result on the expression levels of either viral or other gene expression. Overall, the results are not clearly presented. For example, figure 3 (which is actually a table) presents data, which are uncomprehensible. The legend is poor, there is no explanation of abbreviations, or what the table actually explains or presents. Overall, the present manuscript although interesting, it is not well written nor well presented.
Author Response
Below is our point‐by‐point response to the reviewers’ comments. We thank the reviewers for their constructive suggestions, which have helped us improve the clarity, depth, and presentation of our manuscript. Our responses to each comment are provided below the original remark.
Answers Reviewer 3
In the present work Todorova et al. reported on additional evidence supporting the modern hypothesis regarding the role of persistent HHV infection as a potential etiological co-factor for intraprostatic inflammation, resulting in Prostate Cancer (PCa) tumorigenesis, triggering an immunocompromised tumour profile. In order to evaluate their hypothesis, the authors assessed the PCa tumor microenvironment for impaired immunity, while performing a pilot RNA- expression analysis for PCa-associated immunological factors, some of them including IL1β, IL10, IL 18, TNF-α, TLR4, GATA3, CD68. Although the idea is interesting there are several important aspects that need to be addressed. First of all, the authors report on DNA and RNA extraction from FFPE samples, yet my first concern is that FFPE samples are very hard to obtain RNA from. How did the authors manage to overcome RNA cleavage, after preserning samples in paraffin? The authors mention the evaluation of several genes, but there are no primers mentioned for all these genes. In addition, there should be two different sections; one for DNA, RNA extraction and one for qRT-pCR. Further on, how was viral load estimated? In order to establish a correlation between viral infection and PCa, it is imperative to measure the levels of viral genome in a sample. Yet, there are no data presented for those questions. In particular, the authors did not present any result on the expression levels of either viral or other gene expression. Overall, the results are not clearly presented. For example, figure 3 (which is actually a table) presents data, which are uncomprehensible. The legend is poor, there is no explanation of abbreviations, or what the table actually explains or presents. Overall, the present manuscript although interesting, it is not well written nor well presented.
First of all, the authors report on DNA and RNA extraction from FFPE samples, yet my first concern is that FFPE samples are very hard to obtain RNA from. How did the authors manage to overcome RNA cleavage, after preserving samples in paraffin?
Thank you for this comment. As a powerful source of data, cancer FFPE samples are used for RNA extraction and consequent analyses worldwide. To guarantee the quality of extracted high-molecular weight RNA inputs, all RNA samples were measured concentration and purity by a nanodrop machine. Furthermore, we used of reagents and protocols specifically designed for FFPE tissues to maximize recovery of even fragmented RNA for downstream applications (see Chapter “Material and methods”). Finally, our results are verified based on the internal controls for the quality of the extracted RNA and copy DNA synthesis.
The authors mention the evaluation of several genes, but there are no primers mentioned for all these genes. Yet, there are no data presented for those questions.
We appreciate the question. In our study, the evaluation of gene expression for markers such as IL1β, IL10, IL18, TNF-α, TLR4, GATA3, and CD68 was performed using commercially available diagnostic kits as outlined in the materials and methods section. These kits come with proprietary, pre-validated primer and probe sets that have been optimized for sensitivity and specificity in tissue samples. Because these reagents are standardized and validated by the manufacturers, we did not include the individual primer sequences in the manuscript. The only data we present here is a summary of the inflammatory index via the total expression profile investigated. As indicated, we are planning on investigating the individual factors in subsequent studies.
In addition, there should be two different sections; one for DNA, RNA extraction and one for qRT-pCR. Further on, how was viral load estimated? In order to establish a correlation between viral infection and PCa, it is imperative to measure the levels of viral genome in a sample. In particular, the authors did not present any result on the expression levels of either viral or other gene expression.
Thank you for this comment. The DNA and RNA extraction section is combined into one as we use standardized kits for the extraction procedures and there have been no deviations from the manufacturer’s instructions. There is a subsequent section for the qRT-PCR.
With regard to the viral load, as outlined in the materials and method section, we used kits that target only active HHV infection which do not detect borderline or latent stages of HHV infection, in which case the results correspond to a viral load for an active infection. We are interested in expanding our research in a future study by investigating the levels of viral genome in the samples.
- Overall, the results are not clearly presented. For example, figure 3 (which is actually a table) presents data, which are uncomprehensible. The legend is poor, there is no explanation of abbreviations, or what the table actually explains or presents. Overall, the present manuscript although interesting, it is not well written nor well presented.
Thank you for this observation. Figure 3 illustrates a PCa elevated immunological expression profile. The abbreviations represent the name of the genes, encoding target immunological factors, the elevated inflammation index and the ratios between the factors set in the commercial kit. We also showed the index of B2M=5,6 using as an internal control for RNA extraction and reverse transcription.
If the table is not informative and redundant according the reviewer, it could be eliminated from the manuscript.
We hope that these revisions and clarifications adequately address the reviewers’ concerns and improve the manuscript. Thank you again for your valuable input.
Round 2
Reviewer 2 Report (Previous Reviewer 2)
Comments and Suggestions for Authors
The authors addressed all the comments and the manuscript is suitable for publication
Author Response
Dear Reviewer,
Thank You so much for your positive feedback. Implementing your recommendations significantly improved the quality of the manuscript, for which we thank You sincerely
Reviewer 3 Report (Previous Reviewer 1)
Comments and Suggestions for Authors
The authors have not addressed any of my previous comments. They have responded in their rebuttal letter but no changes were to be seen in their manuscript. In addition, their manuscript lacks data. They report on gene expression studies, yet there are no results and no data to show their findings. For example, I have suggested to change figure 3 to table (which is actually a table) and also make the legend more comprehensible, yet the authors did not perform this relatively simple task. Overall, their work should be significantly improved before publication.
Comments on the Quality of English LanguageProof-read with an English-speaking reviewer.
Author Response
Below is our point‐by‐point response to the reviewers’ comments. We thank the reviewers for their constructive suggestions, which have helped us improve the clarity, depth, and presentation of our manuscript. Our responses to each comment are provided below the original remark.
Answers Reviewer 3
Round 1
In the present work Todorova et al. reported on additional evidence supporting the modern hypothesis regarding the role of persistent HHV infection as a potential etiological co-factor for intraprostatic inflammation, resulting in Prostate Cancer (PCa) tumorigenesis, triggering an immunocompromised tumour profile. In order to evaluate their hypothesis, the authors assessed the PCa tumor microenvironment for impaired immunity, while performing a pilot RNA- expression analysis for PCa-associated immunological factors, some of them including IL1β, IL10, IL 18, TNF-α, TLR4, GATA3, CD68. Although the idea is interesting there are several important aspects that need to be addressed. First of all, the authors report on DNA and RNA extraction from FFPE samples, yet my first concern is that FFPE samples are very hard to obtain RNA from. How did the authors manage to overcome RNA cleavage, after preserning samples in paraffin? The authors mention the evaluation of several genes, but there are no primers mentioned for all these genes. In addition, there should be two different sections; one for DNA, RNA extraction and one for qRT-pCR. Further on, how was viral load estimated? In order to establish a correlation between viral infection and PCa, it is imperative to measure the levels of viral genome in a sample. Yet, there are no data presented for those questions. In particular, the authors did not present any result on the expression levels of either viral or other gene expression. Overall, the results are not clearly presented. For example, figure 3 (which is actually a table) presents data, which are uncomprehensible. The legend is poor, there is no explanation of abbreviations, or what the table actually explains or presents. Overall, the present manuscript although interesting, it is not well written nor well presented.
First of all, the authors report on DNA and RNA extraction from FFPE samples, yet my first concern is that FFPE samples are very hard to obtain RNA from. How did the authors manage to overcome RNA cleavage, after preserving samples in paraffin?
Thank you for this comment. As a powerful source of data, cancer FFPE samples are used for RNA extraction and consequent analyses worldwide. To guarantee the quality of extracted high-molecular weight RNA inputs, all RNA samples were measured concentration and purity by a nanodrop machine. Furthermore, we used of reagents and protocols specifically designed for FFPE tissues to maximize recovery of even fragmented RNA for downstream applications (see Chapter “Material and methods”). Finally, our results are verified based on the internal controls for the quality of the extracted RNA and copy DNA synthesis. This information has been included in the materials and methods section of the manuscript.
The authors mention the evaluation of several genes, but there are no primers mentioned for all these genes. Yet, there are no data presented for those questions.
We appreciate the question. In our study, the evaluation of gene expression for markers such as IL1β, IL10, IL18, TNF-α, TLR4, GATA3, and CD68 was performed using commercially available diagnostic kits as outlined in the materials and methods section. These kits (SensiFAST cDNA Synthesis Kit RXN BIOLINE, ImmunoQuantex C/V REAL-TIME PCR Genotyping Kit and AmpliSens amplification kits, see materials and methods) come with proprietary, pre-validated primer and probe sets that have been optimized for sensitivity and specificity in tissue samples. Because these reagents are standardized and validated by the manufacturers, we did not include the individual primer sequences in the manuscript. We have removed Figure 3, restructuring the results into Table 2 within the manuscript, and we have also performed a statistical analysis to verify the association between HHVs infection and inflammation profile.
In addition, there should be two different sections; one for DNA, RNA extraction and one for qRT-pCR. Further on, how was viral load estimated? In order to establish a correlation between viral infection and PCa, it is imperative to measure the levels of viral genome in a sample. In particular, the authors did not present any result on the expression levels of either viral or other gene expression.
Thank you for this comment. The DNA and RNA extraction section is combined into one as we use standardized kits for the extraction procedures and there have been no deviations from the manufacturer’s instructions. There is a subsequent section for the qRT-PCR, 4.3 Real Time PCR.
With regard to the viral load, as outlined in the materials and method section, we used kits that target only active HHV infection which do not detect borderline or latent stages of HHV infection, in which case the results correspond to a viral load for an active infection. Our aim was to determine whether the presence of an active infection can be an indicator of potential viral contribution to tumorigenesis. The immunological profiling that we conducted (Section 2.3) rovided insights into the local inflammatory environment associated with viral infection. We acknowledge that measuring viral load or gene expression could offer additional quantitative data and strengthen the correlation analysis, and we are interested in exploring this further in subsequent studies.
- Overall, the results are not clearly presented. For example, figure 3 (which is actually a table) presents data, which are uncomprehensible. The legend is poor, there is no explanation of abbreviations, or what the table actually explains or presents. Overall, the present manuscript although interesting, it is not well written nor well presented.
Thank you for your comments which have raised an important point. The revised section 2.3 has been updated to address these concerns. We now clearly explain that the table (renamed Table 2) presents the distribution of inflammation index levels in FFPE PCa samples stratified by HHV infection status. Specifically, it shows:
- For HHVs‐positive PCa samples, 13 out of 34 had an inflammation index below 50%, 3out of 34 had an index above 50%and below 80%, and 18 out of 34 had an index above 80%.
- In contrast, all HHVs‐negative PCa samples (15 out of 15) had an inflammation index below 50%, with none exceeding the 50% or 80% thresholds.
We have also included a detailed legend for Table 2, explaining that the inflammation index is based on immunological factor ratios derived from IL-1β, IL-10, IL-18, TNFα, TLR4, GATA3, and CD68 measurements (with B2M as the housekeeping gene), and that a cut-off value of 50% defines a highly elevated inflammatory profile.
Round 2
The authors have not addressed any of my previous comments. They have responded in their rebuttal letter but no changes were to be seen in their manuscript. In addition, their manuscript lacks data. They report on gene expression studies, yet there are no results and no data to show their findings. For example, I have suggested to change figure 3 to table (which is actually a table) and also make the legend more comprehensible, yet the authors did not perform this relatively simple task. Overall, their work should be significantly improved before publication.
Thank you for your constructive feedback. We have made all efforts to address your comments by revising the manuscript. Specifically, we replaced what was formerly Figure 3 with Table 2, which now displays the distribution of the inflammation index in FFPE PCa samples stratified by HHV infection status. We have also explained that the methodology for calculating the inflammation index (including the cut-off of 50% to define a highly elevated profile) is based on a specific molecular assay kit (ImmunoQuantex C/V REAL-TIME PCR Genotyping Kit) and have performed a statistical analysis to validate our findings (Fisher’s exact test with corresponding p-values).
We acknowledge that while the manuscript now summarizes the gene expression data through the inflammation index and associated statistical comparisons, it does not present the full raw quantitative gene expression values for markers such as IL-1β, IL-10, IL-18, TNFα, TLR4, GATA3, and CD68. We appreciate your observation and understand that including these detailed data would further strengthen our findings. As these are pilot results, our aim was to demonstrate the presence of the elevated immunological profile in the HHVs positive PCa samples and investigate individual gene expression profiles in subsequent studies with specific reagents that would allow us to perform this type of analysis.
Round 3
Reviewer 3 Report (Previous Reviewer 1)
Comments and Suggestions for Authors
The authors have made some revisions in their manuscript, yet there are substantial changes still to be made. The authors still have not presented any data on the experimental procedures concerning RNA expression. Also, the authors have used FFPE samples, for which they should at least show the integrity of RNA (for example by providing supplementary gel electrophoresis showing if RNA is still intact, which is a very simple method). Spectrophotometry shows the concentration, not the quality (even if the ratio is >1.8). Data on gene expression are essential to their investigation. Data on patient demographic and anthropometrics are still missing. Last but not least, there is no evidence for the correlation between viral infection and PCa (at least not in the way data are presented).
The manuscript, still does not have merit for publication.
Comments on the Quality of English LanguageThe manuscript should be proof-read (for example line 309 "...huger...")
Author Response
Thank you for your comments, the manuscript has been proof-read
This manuscript is a resubmission of an earlier submission. The following is a list of the peer review reports and author responses from that submission.
Round 1
Reviewer 1 Report
Comments and Suggestions for Authors
In the present work Todorova et al. performed a herpes virus genotyping attempt in order to investigate the role of viral infection as a poor predictor of prostate cancer with respect to its ontogenesis and progression.
To be honest the whole aim of the present work is totally incomprehensible. To my understanding, the authors attempted to identify a correlation between viral infections and tumor prognosis, diagnosis or even outcome. Yet, the authors presented some descriptive data concerning their patient cohort with no further information on how these correlate to the tumor under investigation, that would be prostate cancer.
There is not information on the patient cohort, apart from a description discriminating the sampling, the authors have no information, no patient data, nothing at all.
Results are presented in a very preliminary manner, which include a pie chart and a Venn diagram. There is no information on correlations and there is no connection on viral infections and tumor characteristics.
Overall, the present work, is not well-written, it appears more of a first draft rather than a research paper ready to be submitted to an international journal. It does not have merit for publication.
Comments on the Quality of English Languageit needs a professional proof-reading.
Reviewer 2 Report
Comments and Suggestions for Authors
In this study, the authors aimed to investigate the role of human herpesviruses (HHVs) in the initiation and progression of prostate cancer (PCa). They assessed the presence of active HHV infections in prostate tissue samples using molecular techniques and found a high frequency of viral infections, particularly EBV, HHV6, and HHV7. The authors suggested that these viruses, especially in co-infection scenarios, contribute to PCa progression through immune dysregulation and inflammatory processes. Furthermore, they explored the immunological implications of HHV infections by analyzing cytokine expression in infected tumor samples.
However, while the study provides interesting data on HHV prevalence and its potential oncogenic role in PCa, several aspects warrant further investigation. The study's sample size, particularly in the control group, appears insufficient for drawing robust conclusions about HHVs’ specific role in prostate tumorigenesis. Additionally, while the findings regarding viral co-infections are intriguing, the underlying mechanistic pathways and their clinical significance require further validation through larger, more diverse cohorts and additional experimental models. Therefore, more comprehensive studies with detailed statistical analysis and clearer mechanistic exploration are needed before the results can be considered sufficient for publication.
Reviewer 3 Report
Comments and Suggestions for Authors
In "Human herpes viruses genotyping and immunological gene expression profiling in prostate cancer with prominent inflammation, Todorova and others investigate the presence of oncogenic viruses like CMV, EBV, and HHV in prostate tissue samples from prostate cancer patients.
This is an interesting topic, but the study is strictly temporary in nature.
Specifically:
1. Only 1, small, cohort is presented, no second independent validation cohort is investigated.
2. Semen fluid does not appear to be a good control sample for healthy tissue.
3. No well-described correlations between virus presence and clinical correlations (patient prognosis, tumor grade, etc) are presented.
4. No clear directions for use in diagnosis and/or treatment are given.
Reviewer 4 Report
Comments and Suggestions for Authors
Human herpes viruses genotyping and immunological gene expression profiling in prostate cancer with prominent inflammation
Introduction, it would help the reader to see more references to review papers and similar work, to better understand the context. What is current opinion on herpes and prostate cancer? See for example:
PMID: 24648964
PMID: 15897985
Line 23, FFPE used here for first time, please change to “Formalin-Fixed Paraffin-Embedded (FFPE) samples”
Line 75, I am confused why this is shown for the FFPE and not the biopsies. The charts are not that large, can two Venn diagrams be presented, Fig 1A and 1B alongside each other.
Also Line 75, the Venn diagram does not make sense. The percentages add up to much more than 100%. Why?
Line 86, a 3D pie chart just makes it harder to read, a 2D pie chart (or a table) conveys the information better. Also, I do not understand what this chart shows that Figure 1 does not?
Line 121, this is not a proper set of results. The actual values of these indicators should be tabulated and presented in some way that is easy to understand. Line 262 (Methods) implies you have measured these things, so present the results in a factual manner.
Discussion, I cannot really tell from the Discussion whether these infections are associated just as much with BPH as with PCA. There is plenty of evidence showing BPH has some of the same pathway changes changes as PCA (inflammation, complement), see for example the below reference. How can the authors be confident herpes is linked to PCA and not just prostate inflammation?
PMID: 38076065
PMID: 17182170
Also in the Discussion it would be better to have more discussion of how the findings compare with existing literature, especially the previous mentioned reviews recommended in the introduction. What is novel about the findings presented here?
Overall, the manuscript requires a lot of work to make the results more clearly presented, and to set these results in context of existing knowledge.